# Potential Risk, Spatial Distribution, and Soil Identification of Potentially Toxic Elements in *Lycium barbarum* L. (Wolfberry) Fruits and Soil System in Ningxia, China

**DOI:** 10.3390/ijerph192316186

**Published:** 2022-12-03

**Authors:** Tongning Zhou, Yan Wang, Jiaqi Qin, Siyuan Zhao, Deyan Cao, Meilin Zhu, Yanxue Jiang

**Affiliations:** 1College of Public Health and Management, Ningxia Medical University, Yinchuan 750004, China; 2College of Basic Medical Sciences, Ningxia Medical University, Yinchuan 750004, China; 3College of Environment and Ecology, Chongqing University, Chongqing 400045, China

**Keywords:** wolfberry–soil system, potentially toxic element, ecological risk, health risk, spatial distribution, source identification

## Abstract

Eight potentially toxic elements (PTEs, including nickel (Ni), copper (Cu), zinc (Zn), arsenic (As), cadmium (Cd), lead (Pb), chromium (Cr), and mercury (Hg)) in *Lycium barbarum* L. (wolfberries) and the associated root soil from a genuine producing area were analyzed. The potential ecological risk of PTEs in the soil and the health risk of PTEs through wolfberry consumption were determined. Geostatistical methods were used to predict the PTE concentrations in the wolfberries and soil. Positive matrix factorization (PMF) was applied to identify the source of PTEs in the soil. The PTE concentrations in the soils were within the standard limits, and Cd in the wolfberries exceeded the standard limit at only one site. The bioconcentration factors (BCF) order for the different PTEs was Cd > Cu > 1 > Zn > Cr > As > Ni > Pb, indicating that Cd and Cu were highly accumulated in wolfberries. The multiple regression models for Ni, Cu, Zn, As, Pb, and Cr concentrations in the wolfberries exhibited good correlations (*p* < 0.1). The ecological risk for Hg in the soil was high, whereas the risks for the remaining PTEs were mostly medium or low. Health risks for inhabitants through wolfberry consumption were not obvious. The spatial distributions of the PTEs in the soil differed from the PTE concentrations in the wolfberries. Source identification results were in the order of natural source (48.2%) > industrial activity source (27.8%) > agricultural activity source (14.5%) > transportation source (9.5%). The present study can guide the site selection of wolfberry cultivation and ensure the safety of wolfberry products when considering PTE contamination.

## 1. Introduction

Studies of potentially toxic elements (PTEs) in soils have attracted the attention of researchers worldwide [1,2]. Human activity (including industrialization, mineral mining, and smelting), as well as the long-term production of sewage sludge, chemical fertilizer use, and pesticide application to soils, has resulted in soil pollution on a global scale [3,4,5].

As of 2018, nearly 20% of the cultivated land in China was polluted by PTEs [6]. PTEs have poor mobility in soil and generally accumulate in the surface layers, as they are mainly combined with soil organic matter [7]. PTEs in the soil can influence the quality of crops, which are consumed by animals or human beings [8]. Therefore, soil pollution by PTEs presents a significant problem for sustainable agriculture, ecological health, and human health [9,10]. Various methods are used to evaluate the pollution of PTEs in soil, and the potential ecological risk indices (RIs) and enrichment factors (EFs) are widely used with many advantages, including their consideration of the synergy, toxic levels, and ecological sensitivities of PTEs [11,12]. It is very useful for policymakers to understand the source of PTEs in soil. A number of receptor models are widely used for source analysis (e.g., principal component analysis, the chemical mass balance method, positive matrix factorization models, and UNMIX models) because they do not require source profiles [13]. Of the many methods, positive matrix factorization (PMF) has been widely applied in many previous studies [14,15]. In the present study, PMF was also adopted to identify PTEs in soil.

Wolfberry (*Lycium barbarum* L.) is a plant belonging to the *Lycium* genus of the Solanaceae family. Wolfberry typically refers to the dried and mature fruit of *L. barbarum* from Ningxia (China) and is the common term for commercial *L. barbarum*, Ningxia *L. barbarum*, Chinese *L. barbarum*, and other *Lycium* species [16]. As a result of intensive studies of wolfberry polysaccharides [17,18], Chinese wolfberry is regarded as an important crop for both medicinal and food uses [16]. However, high levels of PTEs are present in wolfberries compared with the standard limits set by the World Health Organization (WHO), Europe, the International Organization for Standardization (ISO), and China [19,20]. The consumption of wolfberries grown in local fields contaminated with PTEs, therefore, presents a health risk for local inhabitants [21]. Due to increasing global commercialization, further investigation is required if wolfberry cultivation areas increase [22], particularly concerning the prediction of the PTE concentrations in wolfberries using soil indices, which has not been studied previously.

Researchers have investigated the spatial distribution of PTE contamination using geostatistical technology based on geographic information systems (GIS) [23,24,25]. GIS can provide large areas and intuitive results, conduct optimal unbiased interpolations for spatial data, and simulate the discreteness and volatility of the spatial data [26]. Previous studies have generally focused on the PTE concentrations in soils and common crops [27,28]. However, PTEs in wolfberry have rarely been studied.

PTEs can cause acute or chronic harm to humans [29]. For example, Pb can directly cause damage to the human nervous system and can even lead to neonatal congenital mental retardation [30]. Cd is highly toxic and can cause rapid death; in addition, it can accumulate in the kidneys and liver, causing damage to these organs, such as cirrhosis, as well as the digestive system. Cd exposure can also result in the loss of bone calcium via urine, causing osteoporosis [31]. As is highly toxic and can cause rapid death. It is easily deposited in the liver and can lead to cirrhosis; further, As is carcinogenic [32]. Cu, Ni, and Zn are harmful to human health when they exceed safe limits [33,34]. Cr is highly toxic with carcinogenic accumulation and may induce gene mutation in the human body [35]. Hg exposure can induce neurological, nephrological, immunological, reproductive, and even genetic disorders [36]. Eight PTEs (Pb, Cd, Cu, As, Ni, Zn, Cr, Hg) in agricultural soil require detection according to the national standards in China (NY/T 395–2012) [37]. In the Chinese Pharmacopoeia (2015 edition), five PTEs (Pb, Cd, Cu As, and Hg) require detection in Chinese herbal medicines [38]. The concentrations of Cr, Ni, and Zn, which are harmful to human health, also require detection to find the correlation between soil and herbal medicine. Therefore, in the present study, eight PTEs (Pb, Cd, Cu, As, Ni, Zn, Cr, and Hg) were selected, their concentrations and associated risks in soils and wolfberries were evaluated, and the spatial distributions of the PTE concentrations were simulated. In addition, the possible sources of the PTEs in the soils were identified.

## 2. Materials and Methods

### 2.1. Study Area and Sampling Methodology

The Ningxia Hui Autonomous Region is the main Chinese wolfberry production area and the origin of this plant; specifically, Zhongwei is a key planting area [39]. The Shapotou District and Zhongning County in Zhongwei are located at 36°6′–37°50′ N, 104°17′–106°7′ E. As a result of its suitable latitude, long sunshine period, large diurnal temperature range, and abundant water resources, this region has been developed into a high-quality planting area for the last 20 years [40,41]. Therefore, these two areas were selected for this study (Figure 1).

A total of 185 wolfberry samples and associated root soil samples (300 g, 0–20 cm depth) were collected from 37 sites (5 samples in each site) in June 2020 (Figure 1). Wooden tools were used, which can prevent the introduction of PTEs during sampling [42]. The wolfberry and soil samples were stored in polyethylene bags, numbered, and immediately transported to the laboratory.

### 2.2. Laboratory Analysis

The soil samples were air-dried, passed through a 10-mesh (2 mm) nylon sieve, and stirred. The finely ground samples were further ground and passed through a 100-mesh (0.149 mm) nylon sieve [43,44]. The methods for determining PTE and physicochemical properties (pH, organic matter (OM) concentrations, and cation exchange capacities (CEC)) of the soil were in accordance with Chinese standards (Appendix A).

The wolfberry samples were dried in an oven at 60 °C, ground using a high-speed universal disintegrator (FW100), and passed through a 60 mesh (0.25 mm) nylon sieve. An amount of 0.2 g of wolfberry powder was digested with 10 mL of mixed acid (HNO3/HCIO4 = 4:1) overnight. Then, the solution was placed into the digestion instrument (ED54-iTouch) with the following procedure: (i) the temperature was increased to 60 °C and maintained for 20 min; (ii) the temperature was increased to 120 °C and maintained for 30 min; and (iii) the temperature was increased to 180 °C and maintained until the acid solution was entirely eliminated (approximately 6 h). Finally, the solution was cooled to room temperature and diluted using 10 mL ultrapure water.

The elemental composition was determined using inductively coupled plasma optical emission spectrometry (ICP-OES; Agilent Varian710-ES, Santa Clara, CA, USA) [5]. To verify the accuracy and precision of the digestion and analysis procedures, analyses of duplicate samples, blank reagents, and standard reference soil (GBW07419, from the National Research Center for Standards in China) and cabbage (GBW10014) were used in each sample batch. The correlation coefficients of each element were >0.9990. The recoveries of the elements ranged from 75 to 110%, and the relative standard deviation (RSD) values were <5%. 

### 2.3. Bioconcentration Factor

The bioconcentration factor (*BCF*) was determined to evaluate the abilities of the different PTEs to migrate from the soil into the wolfberries. When the *BCF* ≤ 1.00, a plant absorbs but does not accumulate metals. However, if the *BCF* > 1.00, a plant absorbs and also accumulates metals [45]. The *BCF* is given by Equation (1):(1)$$BFC=\frac{Ci}{Csoil}\;$$
where Ci is the concentration of each PTE in wolfberries (mg/kg), and  Csoil is the concentration of the PTE in the soil (mg/kg).

### 2.4. Multiple Linear Regression

Multiple linear regression is a statistical technique used to estimate the relationships between variables. A stepwise regression method was used to perform the multiple linear regression. F-tests were used to assess the model quality. The α value was set to 0.10–0.15 because the number of samples was small [46]. The general formula is given by Equation (2):(2)$$Y=b_{1}X_{1}+b_{2}X_{2}+b_{3}X_{3}+\cdots +b_{x}X_{n}$$
where *Y* is the predicted dependent variable (PTE concentration in wolfberries), $b_{0}$ to $b_{x}$ are partial regression coefficients, and $X_{1}$ to $X_{n}$ are different independent variables (e.g., PTE concentration, pH, OM, and CEC in the soil).

### 2.5. Potential Ecological Risk Index (RI)

The RI method was used to evaluate the PTE pollution in the soil [47]. The method considers not only the PTE concentration in the soil, but also the multi-elemental synergy, PTE toxicities, and environmental sensitivity to PTEs [9]. The RI is calculated using Equations (3) and (4) [12]:(3)$$RI=\sum_{i=1}^{m}E_{r}^{i}$$
(4)$$E_{r}^{i}=T_{r}^{i}*PI=T_{r}^{i}*\frac{Ci}{Bi}$$
where *m* is the number of studied PTEs, $E_{r}^{i}$ is the individual potential ecological risk of element *r*, $T_{r}^{i}$ is the toxicity response coefficient of element *r*, with the values of 40, 10, 5, 1, 5, 30, 5, and 2 for Hg, As, Cu, Zn, Ni, Cd, Pb, and Cr, respectively, *PI* is the pollution index of an individual element, Ci is the actual measurement of the metal element i in the surface soil (mg/kg), and Bi is the reference value, and the background value is applied with the values of 0.02, 11.9, 22.1, 58.8, 36.6, 0.112, 20.6, and 60 mg/kg for Hg, As, Cu, Zn, Ni, Cd, Pb, and Cr, respectively (GB 15618—2018) [48]. The classifications of *RI* and *E*_r_ are presented in Appendix A.

### 2.6. Enrichment Factor (EF)

The EF can be used to determine whether PTEs in the soil are derived from natural or anthropogenic factors. The *EF* is given by Equation (5):(5)$$EF=\frac{\frac{Xi_{sample}}{RE_{sample}}}{\frac{Xi_{background}}{RE_{background}}}$$
where $Xi_{sample}\;$ is the PTE concentration of the soil, $RE_{sample}\;$is the concentration in the reference element (*RE*), $Xi_{background}\;$is the background value for the PTE concentration in the soil, and $RE_{background}\;$ is the background value for the RE concentration [44,49]. Sc, Ti, Zr, V, and Fe are usually used as reference elements [50], and in the present study, Fe (14,509 mg/kg in samples, 26,500 mg/kg of background [51]) was collected as the reference element. The EF degree of contamination is presented in Appendix A.

### 2.7. Human Health Risk Assessment of PTEs 

Among the PTEs considered, Ni, Cu, Zn, As, Cd, Pb, and Cr pose non-carcinogenic health risks through oral exposure. Their risks are given by Equations (6)–(8) [2]:(6)$$HI=\sum_{1}^{n}HQ_{n}$$
(7)HQ=EXPO/RfD
(8)$$EXPO=\frac{C\times DI\times EF\times ED}{BW\times AT}\;$$
where *HQ* is the hazard quotient, *HI* is the hazard index, *EXPO* is the daily exposure to PTEs (mg/(kg·day)), *RfD* is the reference dose (mg/(kg·day)) suggested by the United States Environmental Protection Agency (USEPA) or World Health Organization (WHO), *C* is the PTE concentration in wolfberries (mg/kg), *DI* is the daily intake of wolfberries (dose of 6–15 g/day, dry fruits) suggested by the Chinese Pharmacopoeia (2015 edition) [38] and previous reports [52,53], *EF* is the exposure frequency (assumed to be 90 day/year), *ED* is the exposure duration (year), *BW* is the body weight of the residents (assumed to be 65 kg), and *AT* is the average time (*ED* * 365) [2,54,55].

Through oral exposure, As can also pose carcinogenic health risks (*R*), which can be calculated using Equations (9) and (10) [2]:(9)$$R=SF\times EXPO_{As}$$
(10)$$EXPO_{As}=\frac{c\times DI\times EF\times ED}{BW\times LT}$$
where *SF* is the slope factor with a suggested value of 1.5 mg/[kg·day] [56]. *LT* (lifetime) is the average lifespan of consumers and was considered to be 76.34 years [57].

### 2.8. Spatial Distribution of PTEs

The ordinary kriging (OK) method was used to identify the spatial distribution of PTE concentrations in the soil and wolfberries. Firstly, semivariograms were calculated to describe the structural characteristics, the randomness of PTEs in the soil, and spatial correlation [58]. The general form of a semivariogram is given by Equation (11):(11)$$\;\;\;\gamma \begin{array}{c} {}*\\ \left(h\right) \end{array}=\frac{1}{2N\left(h\right)}\overset{N\left(h\right)}{\underset{i=1}{\sum }}{\left[Z\left(x_{i}\right)-Z\left(x_{i}+h\right)\right]}^{2}$$
where *h* is the spatial separation of two sample points,$\;Z\left(x_{i}\right)\;$is the observed value for the soil sample Z(x) at spatial position *x*_*i*_, $\;Z\left(\;x_{i}+h\;\right)\;$is another observed value for the soil sample Z(x) separated from the original data point by a distance *h*,$\;\;\gamma_{\left(h\right)}^{*}$ is the experimental variation function of the observations separated by a distance *h*, that is, the degree of dissimilarity between points$\;Z\left(x_{i}\right)\;$and$\;Z\left(x_{i}+h\right)$, and N(h ) is the number of sampling point pairs. 

Then, a theoretical model of the semivariogram (e.g., a Gaussian or spherical model) for PTE concentrations in the soil and wolfberries was selected. Finally, OK interpolations were applied to obtain the spatial distribution map in ArcGIS 10.0.

### 2.9. Source Identification

In the present study, EPA PMF 5.0, which was developed by the USEPA, was adopted to identify the source of PTEs in the soil. According to the guidelines, the input data includes PTE concentrations and uncertainties. If the concentration is less than or equal to the method detection limit (*MDL*), the uncertainty (*U_nc_*) is calculated using the following Equation (12):(12)$$U_{nc}=\frac{5}{6}\times MDL$$

If the concentration is greater than the *MDL*, the uncertainty is calculated by Equation (13):(13)$$U_{nc}=\sqrt{{\left(\mathrm{Error}\;\mathrm{fraction}\times C\right)}^{2}+{\left(0.5\times MDL\right)}^{2}}$$

The empirical value of the error fraction is 10% [59].

### 2.10. Statistical Analyses

The means, standard deviations (SDs), and coefficients of variations (CVs) were calculated using Excel 2010. Normal Pearson correlation and multiple linear regression analyses were conducted in SPSS 17. OK interpolations were conducted using ArcGIS 10.0. Source identification was performed in EPA PMF 5.0.

## 3. Results

### 3.1. PTEs in Soil and Wolfberries

The mean PTE concentrations in the soil are shown in Table 1. The Average Ni, Pb, and Hg concentrations (41.0, 23.0, and 0.069 mg/kg, respectively,) were higher than their corresponding background values (36.6, 20.6, and 0.02 mg/kg, respectively) [48]. However, all of the PTE concentrations were lower than their standard limits in China. The CVs of Cd and Hg were higher than 50%, indicating a wide dispersion and strong variability for the two elements.

The PTE concentrations in the wolfberries are shown in Table 2. Hg was not detected in all wolfberry samples. As and Pb were detected in the majority of samples, with detection rates of 59.5% and 52.0%, respectively, and other PTEs were detected in all samples. Compared with the standard limits established by China and the WHO, Cd in one sample exceeded the standard by 2.7% (0.03 mg/kg). Cu, As, Pb, and Hg concentrations in the wolfberries did not exceed the established standards of the Chinese Pharmacopoeia (2015 edition) [38]. As, Cd, Pb, and Hg concentrations in the wolfberries did not exceed the standard limits established by Europe and the ISO. Ni, Zn, and Cr were not compared because no relevant standard limits have been established for these elements in herbal medicine. The BCF values for Cu and Cd were greater than 1, indicating that these two elements were highly enriched in wolfberries.

The results of the Pearson correlation are presented in Appendix A. Cu in wolfberries was significantly positively correlated with soil As and CEC (*p* < 0.05). Zn in wolfberries was significantly positively correlated with CEC (*p* < 0.05). Pb in wolfberries was significantly positively correlated with pH (*p* < 0.05). 

The results of multiple linear regressions are shown in Table 3. The regression equations reflect the relationships between PTE concentrations in wolfberries and soil indices. If the soil indices were provided, the PTEs in wolfberries could be calculated. The relationships between Cu, Zn, Pb, and Ni concentrations in the wolfberries and soil indices were established successfully (*p* < 0.05), indicating that the PTE concentrations in wolfberries can be predicted accurately by soil indices. However, the relationships between As, Cr, and Cd in wolfberries and the soil indices were not well-established, with *p*-values greater than 0.05.

### 3.2. Potential Ecological Risk in Soil 

*E_r_* and *RI* values are presented in Table 4. For *E_r_*, Hg exhibited a high potential ecological risk level (80 < *E_r_* < 160), while the other six PTEs exhibited low potential ecological risk levels (Zn < Cu < Cr < Ni < Pb < As < Cd < 40). For the average *RI*, the soils were evaluated as having medium potential ecological risk levels (150 < *RI* < 300), with Hg having a contribution rate of 86.5%.

The *EF* values of the soils are listed in Table 4. The average *EF* values indicate that all of the PTEs were classified as nonpolluting or minimally polluting (Cd < Cu < As < Zn < Cr < 2 < Ni < Pb < Hg), indicating that Ni, Pb, and Hg are heavily influenced by human activities.

### 3.3. Potential Health Risk in Wolfberries

The non-carcinogenic risk of individual elements (HQ) was 0.00749, 0.0371, 0.0111, 0.116, 0.0174, 0.0168, and 0.149 for Ni, Cu, Zn, As, Cd, Pb, and Cr, respectively. The non-carcinogenic risk of the combined elements (*HI*) was 0.355. The *HQ* (Ni < Zn < Pb < Cd < Cu < As < Cr < 1) and *HI* (<1) values of the PTEs indicate that no obvious non-carcinogenic risk is present. The contribution rates of Ni, Cu, Zn, As, Cd, Pb, and Cr to the *HI* were 2.11%, 10.5%, 3.13%, 32.7%, 4.90%, 4.72%, and 41.9%, respectively. Cr and As were the main elements in the wolfberries that cause a non-carcinogenic risk. The carcinogenic risk caused by As (R) was 2.01 × 10^−7^, which was lower than 10^−6^, indicating that the carcinogenic risk through wolfberry consumption was negligible. 

### 3.4. Spatial Distribution of PTEs in Soil and Wolfberries

Details of the semivariogram results of PTEs in the soil are shown in Appendix A. In the spatial distribution of soil PTEs (Figure 2), the east–west trend is strong, whereas the north–south trend is weak. Ni and Cr, as well as Zn and Pb, had similar spatial distribution tendencies. In the western part of the study area, the Zn, As, and Pb concentrations were relatively low in the soil, whereas the Cr, Hg, and Ni concentrations were relatively high. In the central part of the study area, all of the PTEs exhibited peak values. In this area, the highest Cu concentration was located next to the lowest concentration, indicating that soil Cu migration is not significant. In the eastern part of the study area, the Ni and Cr concentrations both exhibited high values, and the Cu concentration increased compared with the central part of the study area.

The results of the semivariograms for PTEs in wolfberries are shown in Appendix A. PTE concentrations in the wolfberries were predicted using soil indices by using the linear regression equation shown in Table 5. Figure 3 shows the results of the OK interpolation of wolfberry PTE concentrations. The PTE concentrations in the wolfberries were generally high in the western part of the study area and low in the eastern part of the study area, which is not consistent with the distribution of PTEs in the soil.

### 3.5. PMF Source Identification

Four factors were identified using EPA PMF 5.0, and factor profiles (% of species total), as illustrated in Figure 4. The four factors were the main sources of PTEs in the soil, with contribution rates of 9.5%, 27.8%, 14.5%, and 48.2%. 

Factor 1 was mainly dominated by Cd (99.8%), followed by Pb (18.1%), Zn (17.0%), and Ni (14.1%). Previous studies have found that Cd and Pb are the main indicators for transportation, including vehicle fuel combustion, vehicle engines, and tire friction [60]. In addition, Ni and Zn were also found to exist in automobile exhausts which can accumulate in cultivated soil through air–dust adsorption and atmospheric deposition [61]. Therefore, it was speculated that factor 1 represents the transportation source. 

Factor 2 was dominated by Cr (57.3%), Hg (26.0%), As (21.2%), and Cu (20.5%). Although Cr, Ni, and Cu were found to be strongly related to the soil parent material in a previous study [62], Hg was also a main contributor to the factor. The CV of Hg in the soil was 52%, indicating that Hg levels are influenced by human activities. Hence, compared with factor 4, factor 2 was not a natural source. Hg and As were found to be related to fossil fuel combustion in a previous study [60]. In addition, the high value of As in the soil appeared in the western part of the study area where Zhongning Ningxia Industrial Park is located. Therefore, the presumed factor 2 was the industrial activity source.

Factor 3 was mainly dominated by Hg (74.0%). The CV of Hg in the soil was 52%, indicating that Hg levels are influenced by human activities. A previous report found that Hg is an important component of pesticides and fertilizers [63]. In addition, in the present study, the high value of Hg in the soil appeared in the central and western parts of the study area, which are located in the cultivated region. Therefore, factor 3 was considered to be the agricultural activity source.

Factor 4 was mainly dominated by Cu (53.1%), As (51.7%), Ni (51.2%), Zn (50.9%), Pb (46.2%), and Cr (42.2%). The CVs of these metals were less than 40%, indicating that these four metals in the soil are not obviously affected by human activities. Furthermore, the Cu, As, Zn, and Cr concentrations in the soil were below the background values for the soil in Ningxia, and Cr, Ni, and Cu were found to be strongly related to the soil parent material in a previous study [60,62]. Therefore, factor 4 was presumed to be the natural source.

In summary, the four factors followed the order of natural source (48.2%) > industrial activity source (27.8%) > agricultural activity source (14.5%) > transportation source (9.5%).

## 4. Discussion

In this study, the PTE concentrations in the wolfberries were relatively low, which is consistent with previous findings. Kai et al. [64] also found that As, Hg, Pb, Cd, Cr, and Ni concentrations did not exceed the standard limits for wolfberries. However, Xiao et al. [65] studied the PTE concentrations in wolfberries from Qinghai Province and found that concentrations of Pb (2.22%) and Cd (35.6%) in the wolfberries exceeded the standard limits. Xiao et al. [66] studied wolfberry plantations in Qaidam Basin and found that wolfberries were subject to excessive Cd exposure. Qi et al. [20] evaluated Ningxia wolfberries from six different producing areas in China (Ningxia, Inner Mongolia, Hebei, Xinjiang, Gansu, and Qinghai) and found that no PTE concentrations exceeded the standard limits. Therefore, the PTE concentrations in wolfberries from different producing areas vary widely.

The Cd and Cu concentrations in the wolfberries were far greater than those of the soil, indicating that absorption and accumulation occur. Qi et al. [20] found that the BCF value for Cd in wolfberries (0.318–0.903) was also larger than that of other PTEs, while the average BCF values for Ni, As, and Pb were less than 0.1, which is consistent with the results of this study. Petukhov [45] found that the BCF of Cu in Chinese herbal medicines is generally greater than 1 (e.g., chamomile has a BCF of 4.67), and the BCF of Cd was also higher than 1. However, the BCFs of Cu and Cd in vegetables (e.g., tomato) [67] and crops (e.g., rice) [68] were much lower than 1. It can be assumed that Cu and Cd enrichment in Chinese herbal medicines is greater than that in other crops, although this requires further study.

In this study, multivariate linear relationships between the PTE concentrations in the wolfberries and soil indices were established. This method has also been applied to other crops. Chen et al. [58] successfully fitted a linear regression equation for the As concentrations in five crops and root zone soil indices; however, good fits were only obtained for corn and red jujube. McBride [69] obtained an equation where the soil Cd concentrations and pH significantly affected the Cd concentrations in lettuce, beet, and maize leaves (*p* < 0.05). Zhu et al. [70] also obtained a regression equation for the Cd concentrations and root zone soil indices (*p* < 0.01) in *Panax Notoginseng*. These previous studies have shown the applicability and universality of this method; however, in the present study, As, Cr, and Cd concentrations in wolfberries and soil indices did not correlate well (*p* > 0.05), possibly because of the insufficient samples and soil indices. 

In the east–west direction, the PTE concentrations in the wolfberries were high in the west, while the PTE concentrations in the soil were high in the east. It is assumed that the PTE concentrations in the wolfberries were more likely related to the physicochemical properties of the soil. In the correlation analysis, the concentrations of Ni, Cu, Zn, As, Cd, and Pb in the wolfberries were all positively correlated with OM or pH, which can improve our hypothesis. Previous studies also support our findings concerning the pH results. Wang et al. [40] found that PTE adsorption by wolfberries increased with increasing soil pH because a high soil pH can result in decreases in active PTE concentrations in the soil, which can be easily absorbed by plants. However, previous reports are inconsistent with our results for OM. McBride [67] reported that OM in the soil increased the absorption of PTEs by crops because OM contains a large number of functional groups that can combine with active PTEs in the soil and reduce the availability of metals. Therefore, rather than PTEs in the soil, more attention should be paid to the physicochemical properties of the soil. 

In previous reports, the spatial distribution of PTEs in crops has been studied, using the data for PTE concentrations in crops directly [71,72]. However, in the present study, the PTE concentrations in wolfberries predicted by soil indices were used to make a spatial distribution map. The advantages of this method are, firstly, that the soil in a certain range of space is continuous while wolfberries are discontinuous, which is not suitable for statistical analysis [73], and secondly, it helps to select safe areas for wolfberry planting when considering PTE contamination. Due to the limitation of sample size in this study, the prediction accuracy in the multiple linear regression models for some metals is not ideal, but it provides a preferable method.

## 5. Conclusions

The average soil PTE concentrations did not exceed the national standards. Hg in the soil was the most important polluting element, while the other PTEs presented low ecological risks. Cd levels in one wolfberry sample exceeded the standards for herbal medicine while the levels of other metals in the wolfberries were safe. The health risks to inhabitants through wolfberry consumption were not obvious, indicating that wolfberries can be safely consumed. The distribution of different PTEs in the soil was relatively higher in the eastern and central parts of the study area. However, the high PTE concentrations in wolfberries occurred in the western part of the study area. The source of the PTEs in the soil was mainly from nature (48.2%), industrial activities (27.8%), agricultural activities (14.5%), and transportation (9.5%). In summary, these results can provide a method for predicting the spatial distributions of PTEs in wolfberries and guarantee their quality.

## Figures and Tables

**Figure 1 ijerph-19-16186-f001:**
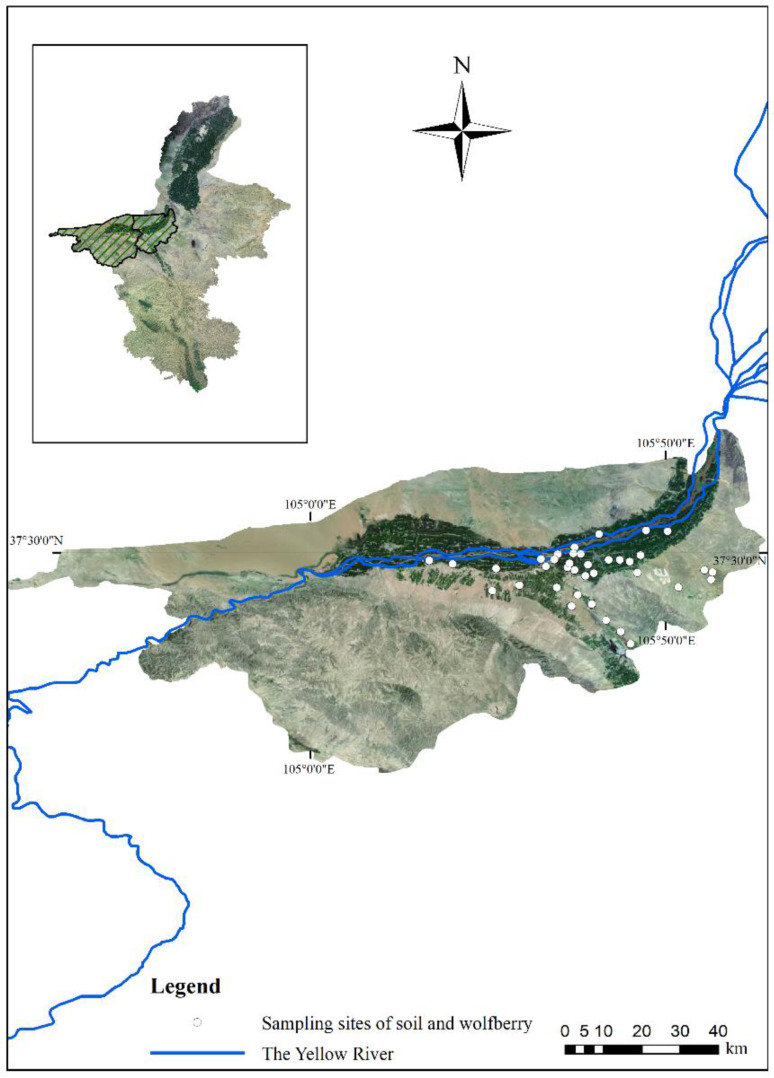
Geomorphology of the study area and sampling site distribution. Note: The east part of the study area is a cultivated plain area, and wolfberries are mainly cultivated in the dark green area along the Yellow River. The western part of the study area is a mountainous, desert area and is not suitable for wolfberry cultivation. All figures in this study are based on the GCS_Krasovsky_1940 coordinate system and D_Krasovsky_1940 datum plane.

**Figure 2 ijerph-19-16186-f002:**
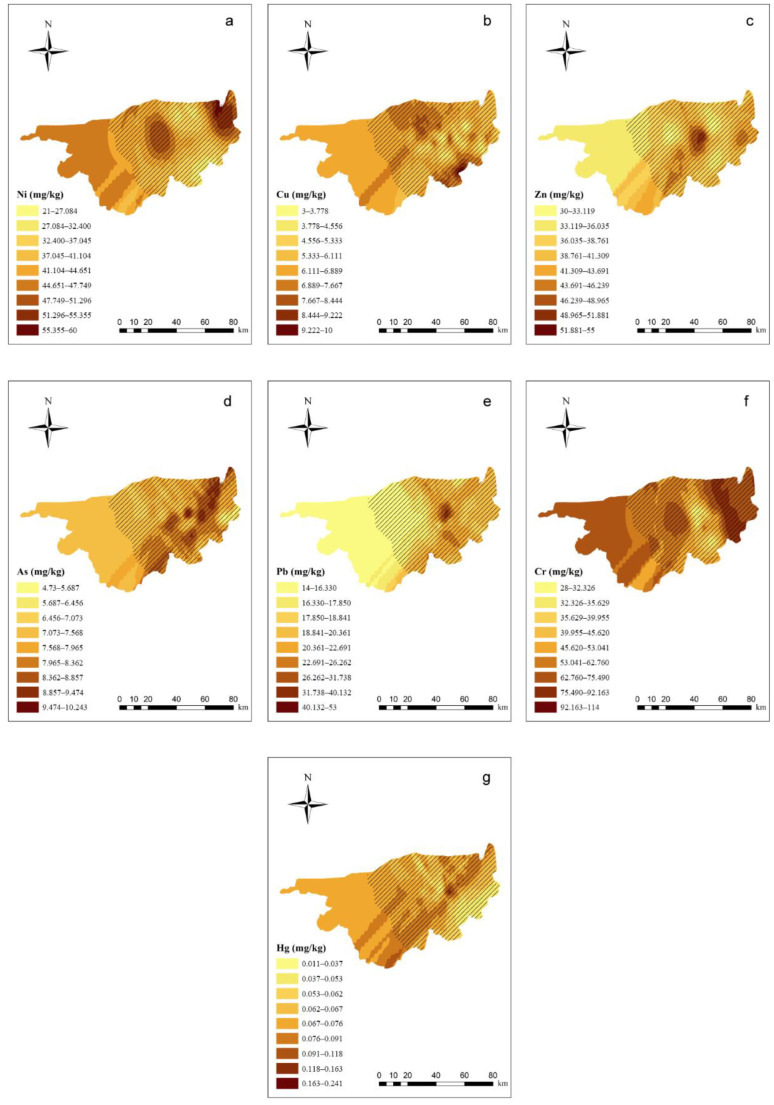
Ordinary kriging distribution charts for the PTE concentrations in the soil of the study area (ArcGIS 10.0). Note: Subgraphs (**a**–**g**) are the distribution charts for the concentrations of Ni, Cu, Zn, As, Pb, Cr, and Hg in the soil. Different color depths represent different concentration ranges, and the deeper the range, the higher the concentration. 
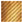
 means the main planting area for wolfberries.

**Figure 3 ijerph-19-16186-f003:**
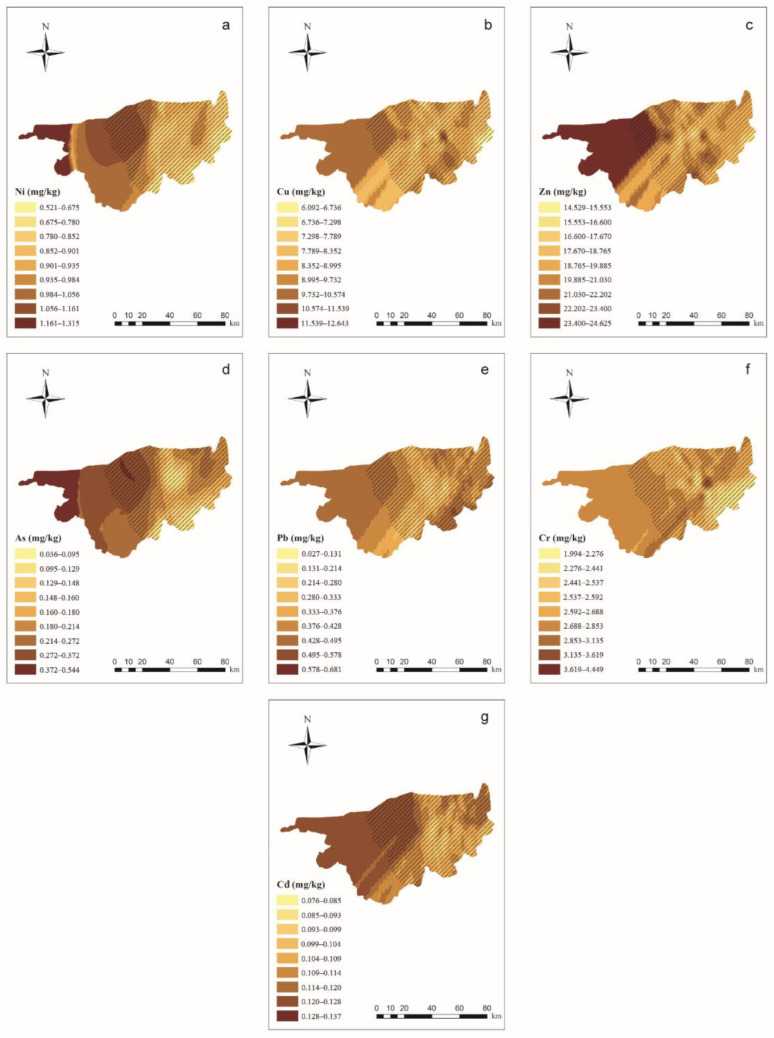
Ordinary kriging distribution charts for wolfberry PTE concentrations in the study area (ArcGIS 10.0). Note: Subgraphs (**a**–**g**) are the distribution charts for the concentrations of Ni, Cu, Zn, As, Pb, Cr, and Cd in wolfberries. Different color depths represent different concentration ranges, and the deeper the range, the higher the concentration. 
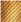
 means the main planting area for wolfberries.

**Figure 4 ijerph-19-16186-f004:**
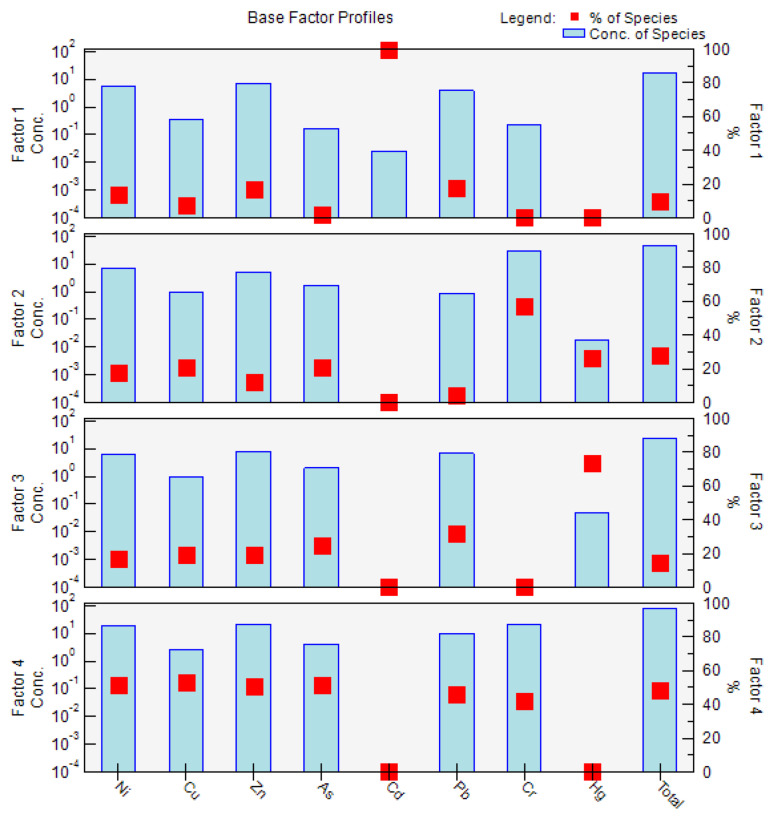
The source apportionment and source profiles for the PTE concentrations in the soil were obtained from positive matrix factorization (PMF). Notes: the red dot indicates the percentage (%) of the species, and the light blue bar indicates the concentration of the species.

**Table 1 ijerph-19-16186-t001:** PTE concentrations in the soil.

	Ni	Cu	Zn	As	Cd	Pb	Cr	Hg
Detection rate (%)	100	100	100	100	94.6	100	100	100
Mean ± SD (mg·kg^−1^)	41.0 ± 9.42	5.27 ± 1.61	42.5 ± 6.21	8.36 ± 1.59	0.0260 ± 0.0240	23.0 ± 9.00	52.9 ± 20.5	0.0690 ± 0.0360
Min (mg·kg^−1^)	21.0	3.00	30.0	4.73	0	14.0	28.0	0.0108
Max (mg·kg^−1^)	60.0	10.0	55.0	11.2	0.130	53.0	114	0.241
Coefficient of variation (%)	23.0	30.6	14.6	19.1	92.7	39.1	38.7	52.0
^a^ Background value (mg·kg^−1^)	36.6	22.1	58.8	11.9	0.112	20.6	60.0	0.0200
^a^ Limited standard value (mg·kg^−1^)	190	200	300	20.0	0.800	240	350	1.00

^a^ From Soil environmental quality risk control standard for the soil contamination of agricultural land (GB 15618—2018) in China.

**Table 2 ijerph-19-16186-t002:** PTE concentrations in wolfberries.

	Ni	Cu	Zn	As	Cd	Pb	Cr	Hg
Mean ± SD (mg·kg^−1^)	0.880 ± 0.440	8.70 ± 2.70	19.6 ± 6.41	0.200 ± 0.230	0.100 ± 0.0700	0.350 ± 0.270	2.62 ± 1.20	^e^ u
Max (mg·kg^−1^)	2.52	14.5	35.4	0.810	0.350	0.960	5.32	u
Min (mg·kg^−1^)	0.210	2.29	11.0	0	0.0300	0	0.170	u
Detection rate (%)	100	100	100	59.5	100	92.0	100	u
^a^ Limited standard A (mg·kg^−1^)	^f^ -	20.0	-	2.00	0.300	5.00	-	0.20
^b^ Limited standard B (mg·kg^−1^)	-	-	-	-	0.300	10.0	-	-
^c^ Limited standard C (mg·kg^−1^)	-	-	-	-	1.00	5.00	-	0.100
^d^ Limited standard D (mg·kg^−1^)	-	-	-	4.00	2.00	10.0	-	3.00
Bioconcentration factor	0.0230	1.76	0.475	0.0260	5.82	0.0170	0.0560	u

^a^ From Chinese Pharmacopoeia (Chinese Pharmacopoeia Commission, 2015) and “Green standards of medicinal plants and preparations for foreign trade and economy” (Ministry of Foreign Trade and Economic Cooperation, 2005). ^b^ From Quality Control Methods for Medicinal Plant Materials Revised Draft Update (2005) for WHO. ^c^ From European Pharmacopoeia. ^d^ From ISO 18664:2015 (Traditional Chinese Medicine—Determination of heavy metals in herbal medicines used in Traditional Chinese Medicine). ^e^ u indicates that it is not detected. ^f^—indicates that the limit standard is not specified.

**Table 3 ijerph-19-16186-t003:** Multiple linear regression equations between PTEs concentration in wolfberries and soil indices.

	^a^ Regression Equation	F	P	r
Ni	−0.271 + 0.028 OM + 0.072 C_As_	entry 0.15 removal 0.2	0.048	0.405
Cu	−13.978 + 1.496 CEC + 0.545 C_Cu_ + 0.15 OM + 0.486 C_As_	entry 0.15 removal 0.2	0.002	0.629
Zn	−17.283 + 3.505 CEC + 0.349 OM	entry 0.15 removal 0.2	0.028	0.436
As	0.041 + 0.012 OM +0.007 C_Ni_ − 0.009 C_Zn_	entry 0.15 removal 0.2	0.053	0.453
Cd	−0.413 + 0.066 pH	entry 0.2 removal 0.25	0.186	0.222
Pb	−3.393 + 0.556 pH − 0.012 C_Ni_ − 1.869 C_Hg_	entry 0.15 removal 0.2	0.002	0.598
Cr	1.879 + 10.664 C_Hg_	entry 0.15 removal 0.2	0.053	0.320

^a^ C_As_, C_Cu_, C_Ni_, C_Zn,_ and C_Hg_ are the concentrations of As, Cu, Ni, Zn, and Hg in the soil. CEC is the cation exchange capacity of the soil. OM is the organic matter of the soil. pH is the pH value of the soil.

**Table 4 ijerph-19-16186-t004:** Potential ecological risks in soils.

	Ni	Cu	Zn	As	Cd	Pb	Cr	Hg	Total
Potential ecological risk	5.33	1.19	0.722	7.02	7.02	5.58	1.76	^a^ 139	^b^ 167
Enrichment factor	2.04	0.436	1.32	1.28	0.428	2.04	1.61	6.33	-

^a^ indicates that the E_Hg_ in soil samples shows heavy risk (80 ≤ *E_r_* < 160). ^b^ indicates that the *RI* of combined metals shows medium risk (150 ≤ *RI* < 300).

**Table 5 ijerph-19-16186-t005:** Multiple linear regression equations between PTE concentrations in wolfberries and soil indices.

	^a^ Regression Equation	F	P	r
Ni	−0.271 + 0.028 OM + 0.072 C_As_	entry 0.15 removal 0.2	0.048	0.405
Cu	−13.978 + 1.496 CEC + 0.545 C_Cu_ + 0.15 OM + 0.486 C_As_	entry 0.15 removal 0.2	0.002	0.629
Zn	−17.283 + 3.505 CEC + 0.349 OM	entry 0.15 removal 0.2	0.028	0.436
As	0.041 + 0.012 OM +0.007 C_Ni_ − 0.009 C_Zn_	entry 0.15 removal 0.2	0.053	0.453
Cd	−0.413 + 0.066 pH	entry 0.2 removal 0.25	0.186	0.222
Pb	−3.393 + 0.556 pH − 0.012 C_Ni_ − 1.869 C_Hg_	entry 0.15 removal 0.2	0.002	0.598
Cr	1.879 + 10.664 C_Hg_	entry 0.15 removal 0.2	0.053	0.320

^a^ C_As_, C_Cu_, C_Ni_, C_Zn,_ and C_Hg_ are the concentrations of As, Cu, Ni, Zn, and Hg in the soil. CEC is the cation exchange capacity of the soil. OM is the organic matter of the soil. pH is the pH value of the soil.

## Data Availability

The datasets used and/or analyzed during the current study are available from the corresponding author upon reasonable request.

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
