# Peer review of "Potential Risk, Spatial Distribution, and Soil Identification of Potentially Toxic Elements in Lycium barbarum L. (Wolfberry) Fruits and Soil System in Ningxia, China"

_ijerph, 2022, doi:10.3390/ijerph192316186_

Round 1

Reviewer 1 Report

1. In the present manuscript, the authors studied the risk, spatial distribution, and soil identification of the potentially toxic elements in the Lycium barbarum with the soil system in Ningxia of China. The study has been systematically presented and the discussion is also well done. However, the language needs a thorough revision in terms of good expressions. 

2. It is also suggested that the authors need to explain the toxicant metals considered in this study. On what basis they selected and what are their toxicity to the soil and human health?

3. The authors need to improve the research hypothesis and novelty of the study. 

4. To discuss the toxicity of the concerned metals, the authors may refer and cite 10.1016/j.chemosphere.2021.131978; 10.1016/j.seppur.2018.09.050; 10.2166/wst.2018.226

5. What is the overall accuracy, calculated from the stratified random points? Was there a problem of low spatial resolution images, if yes how that concern was taken care of?

Reviewer 2 Report

The work is interesting and well-designed, but its realization raises some questions:

·      Please consider changing the title: …….in the Lycium barbarum L. (wolfberry) fruits……..

·      Is Lycium barbarum L. (wolfberry) classified by the Authors as a medicinal plant or food? Please provide an adequate justification (dose of 6-15 g (dry fruits?) taken two or three times daily, lines 157-158).

·      Do the samples of wolfberries come from crops intended for food production or from medical plant crops intended for drug production?

·      Which soil grain fraction was used to determine the background values and limit values of PTEs in soils? Please list available references.

·      The large difference between the iron concentration in the soil samples (14509 mg/kg) and the background concentration (2650 mg/kg) probably reflects the completely different geochemistry of these soils.

·      Line 148: Potential health risk of wolfberries. Please consider “Human health risk assessment of PTEs” (Other contaminants may be also present in wolfberries)

Reviewer 3 Report

This manuscript performed a target analysis on the Ni, Cu, Zn, As, Cd, Pb, Cr and Hg in wolfberries and the soil, reporting the concentration of the potentially toxic elements (PTEs), and calculating various indicators to speculate the spatial distribution and possible sources of the PTEs. However, sufficient discussion is lack on the results calculated by models or equations. Inconsistent inferences can be found in the manuscript. Besides, the quality control was lack in laboratory analysis. The following are my specific comments.

1. lines 85-86: Considering the description of figure 2 and 3, whether it is better to be divided by east and west?

2. lines 89-105: Provide the QA/QC in the part of laboratory analysis.

3. lines 232: Describe it more clearly that how to predicted PTE content in wolfberries by soil indices. This inference was in contradiction with lines 272-278.

4. Figure 2 and 3: The subheadings of the sub-graph need to be commented in the title.

5. Can the value of coefficient of variation (CV) be used to reflect the impact of human activity?

6. Lines 244-246: The inference that PTEs were not much influenced by human activities was not consistent with lines 284-316, where the human activity source account for more than 50% of the total source.

Round 2

Reviewer 1 Report

The revised version of the manuscript can be accepted for publication.

Reviewer 3 Report

I recommend the manuscript can be published in the present form.